# Impact of Biologics on Comorbidities in Patients with Psoriasis or Psoriatic Arthritis

**DOI:** 10.3390/biomedicines13092219

**Published:** 2025-09-10

**Authors:** Sang-Hoon Lee, Solam Lee, Hee Seok Seo, Sang Baek Koh, Minseob Eom, Seung-Phil Hong

**Affiliations:** 1Department of Dermatology, Yonsei University Wonju College of Medicine, Wonju 26426, Republic of Korea; epih92@yonsei.ac.kr (S.-H.L.);; 2Department of Pathology, Yonsei University Wonju College of Medicine, Wonju 26426, Republic of Korea; 3Department of Preventive Medicine, Yonsei University Wonju College of Medicine, Wonju 26426, Republic of Korea

**Keywords:** psoriasis, psoriatic arthritis, biologics, immunotherapy, comorbidity, risk

## Abstract

**Background/Objectives**: Psoriasis and psoriatic arthritis are associated with various comorbidities, particularly cardiovascular conditions. Although biologics are increasingly used to manage moderate-to-severe disease, their effect on comorbidity risk remains unclear. This study aimed to assess the association between biologics and the risk of comorbid diseases compared to conventional systemic immunosuppressants. **Methods**: A retrospective cohort study was conducted using the Korean National Health Insurance Service database from 2002 to 2021. Patients with a principal diagnosis of psoriasis or psoriatic arthritis were included. Overall, 8173 biologics users (TNF-α, IL-12/23, IL-23, or IL-17 inhibitors) were compared to 41,598 patients treated exclusively with cyclosporine A or methotrexate. Adjusted hazard ratios (aHRs) for incident comorbid diseases were calculated using Cox proportional hazard models, with follow-up through 31 December 2021. **Results**: Biologics use was associated with a decreased risk of rheumatoid arthritis (aHR, 0.37; 95% CI, 0.17–0.79), mood disorders (aHR, 0.72; 95% CI, 0.53–0.97), and solid tumors (aHR, 0.63; 95% CI, 0.47–0.84). Subgroup analyses revealed that IL-23 inhibitors were linked to reduced risk of solid tumors (aHR, 0.31; 95% CI, 0.12–0.83), whereas IL-17 inhibitors were associated with increased risk of chronic obstructive pulmonary disease (aHR, 2.96; 95% CI, 1.08–8.14). No significant differences were found for major cardiovascular events. **Conclusions**: Biologics appear relatively safe with respect to cardiovascular disease and may reduce the risk of certain comorbidities such as mood disorders and solid tumors in patients with psoriasis or psoriatic arthritis. Clinicians should consider comorbidity profiles when selecting biologic agents for individual patients.

## 1. Introduction

Psoriasis (PsO) and psoriatic arthritis (PsA) are chronic, immune-mediated, multisystem diseases characterized by erythematous and scaly skin lesions and joint inflammation, respectively [1,2,3]. PsO affects approximately 2–3% of the global population, while PsA develops in up to 30% of individuals with PsO [2,4,5]. In South Korea, the prevalence of PsO is estimated at approximately 0.5%, and recent trends indicate a steady increase in its incidence [6]. The systemic inflammatory nature of these diseases often predisposes patients to a wide range of comorbidities [3,7,8]. Multiple studies have reported increased risks of cardiovascular (CV) disease, metabolic syndrome, chronic kidney disease (CKD), inflammatory bowel disease (IBD), rheumatoid arthritis (RA), depression, and some malignancies among individuals with PsO and PsA [9,10,11,12]. These comorbidities significantly affect the overall morbidity and quality of life [13].

The treatment landscape for PsO and PsA has advanced significantly with the introduction of biologics, starting with tumor necrosis factor (TNF)-α inhibitors and subsequently expanding to include interleukin (IL)-17 and IL-23 inhibitors [2,14]. These agents, targeting specific immune pathways, revolutionized disease management by providing substantial clinical improvements and potentially impacting associated comorbidities [14,15,16,17]. However, most previous studies on the relationship between biologics and comorbidities are derived from clinical trials [18], which are limited by small sample sizes and the assessment of only a limited range of comorbidities [19].

Although concerns about increased CV disease risk have been raised, as evidenced by the withdrawal of the IL-12/23 blocker Briakinumab from the market due to ischemic CV safety concerns [20,21], the precise effect of biologics on comorbid conditions remains unclear. Given these uncertainties, this study aims to evaluate the impact of biologics currently used on comorbid diseases in patients with PsO or PsA compared with that of conventional systemic immunosuppressants such as Cyclosporine A (CsA) and methotrexate (MTX).

## 2. Materials and Methods

### 2.1. Data Source

This retrospective cohort study was conducted in line with the Strengthening the Reporting of Observational Studies in Epidemiology (STROBE) guidelines and was approved by the Institutional Review Board of Yonsei University Wonju College of Medicine (approval no. CR321372). The requirement for informed consent was waived owing to the anonymized nature of the dataset. This study employed data from the Korean National Health Insurance Service (NHIS) database, covering more than 99% of the South Korean population. Since 2002, the database has included comprehensive information on healthcare utilization, diagnoses, and prescription records, as well as some data on health screenings and socioeconomic status. Data from general health examinations and lifestyle surveys conducted during recommended annual or biennial check-ups for employees, household heads, and individuals aged 40 years and older were incorporated.

### 2.2. Study Design and Patients

Patients with PsO or PsA were defined as those with at least one documented visit with the International Statistical Classification of Diseases (10th Revision) code of L40.x as a principal diagnosis between 2002 and 2021. The biologics cohort included patients who received at least one dose of biologic agents classified as TNF-α inhibitors (adalimumab, etanercept, infliximab, and golimumab), IL-12/23 inhibitors (ustekinumab), IL-23 inhibitors (guselkumab and risankizumab), or IL-17 inhibitors (secukinumab and ixekizumab). The index date was defined as the date of the first biologics prescription. Moreover, patients switching between biologic classes were censored at the switch date, with follow-up based on the initially administered class.

As the incidence of comorbid diseases in PsO can be influenced by the severity of PsO [22], the control cohort comprised individuals treated with conventional systemic immunosuppressants (CsA or MTX) without biologics. The index date for patients in the control group was the first prescription date for conventional systemic immunosuppressants. Follow-up for both cohorts began at the index date and ended at the observation period’s conclusion (31 December 2021).

### 2.3. Outcomes

This study evaluated the prevalence and risk of incident comorbid diseases in the biologics cohort compared with those in the control cohort. The primary outcome of interest was the incidence of the following comorbid diseases until 31 December 2021 (Appendix A): major cardiovascular event (MACE), myocardial infarction (MI), stroke, congestive heart failure, atrial fibrillation, transient ischemic attack, peripheral arterial disease, atherosclerosis, rheumatoid arthritis (RA), inflammatory bowel disease (IBD), vasculopathy, immunobullous disease, tuberculosis (TB), hepatitis B virus (HBV) infection, hepatitis C virus (HCV) infection, human immunodeficiency virus infection, herpes zoster, mood disorder, anxiety disorder, dementia, lymphoma, leukemia, solid tumor, non-melanoma skin cancer (NMSC), malignant melanoma, asthma, chronic obstructive pulmonary disease (COPD), and chronic kidney disease (CKD). At least three clinic visits for each outcome with MACE as the principal diagnosis were considered an occurrence of the disease. MACE was defined as having at least one hospitalization with MI or stroke as the principal diagnosis.

### 2.4. Covariates

Demographic, socioeconomic, and lifestyle data were obtained from the NHIS database. Additionally, data on CV risk factors were extracted from the NHIS general health examination database. Patients were categorized into high-risk and low-risk groups based on CV risk factors. The high-risk group was defined as patients with two or more of the following factors: age (over 50 and 60 years for men and women, respectively), hypertension (HTN), diabetes mellitus, dyslipidemia, smoking (current smoker), and obesity (body mass index ≥ 25 kg/m^2^).

### 2.5. Statistical Analysis

Baseline characteristics are summarized as means with standard deviations for continuous variables and frequencies with percentages for categorical variables. Differences between groups were assessed using the *t*-test for continuous variables and the χ^2^ test for categorical variables. The prevalence of comorbid diseases at the index date was evaluated, and adjusted odds ratios (aORs) for the biologics cohort compared to the control group were calculated using multivariable logistic regression analyses.

Incidence rates are expressed as events per 100,000 person-years, and multivariable Cox proportional hazard models were used to estimate adjusted hazard ratios (aHRs) and 95% confidence intervals (CIs). Covariates included age, sex, insurance type, income level, and geographic location. Subgroup analyses explored potential differences by class of biologics and/or CV risk factors. Given the known CV protective association with the long-term use of TNF-α inhibitors and the current widespread use of IL inhibitors in PsO [2,23,24], the biologics cohort was divided into TNF-α inhibitor and IL inhibitor cohorts for this analysis. Furthermore, to differentiate our study from previous claims database studies [25,26], we assessed the risk for each individual CV disease. Analyses by biologics class were conducted on the entire cohort, while analyses based on CV risk factors were restricted to patients with general health examination data available.

All statistical analyses were conducted using SAS statistical software (version 9.4; SAS Institute Inc., Cary, NC, USA) and R statistical software (version 3.6.3; R Foundation for Statistical Computing, Vienna, Austria). A two-sided significance level of 5% was applied for all tests.

## 3. Results

### 3.1. Study Population

The analysis included 8173 biologics-treated patients and 41,598 controls (Figure 1). The demographics and baseline characteristics of the entire study population are summarized in Appendix A. Moreover, the details of the subgroup who underwent a general health examination, including the number of CV risk factors, are presented in Appendix A. The biologics cohort had more patients with PsA (11.9% vs. 0.3%, *p* < 0.001), men (66.2% vs. 57.6%, *p* < 0.001), and a younger mean age (49.1 vs. 52.7 years, *p* < 0.001) than the control group. Furthermore, the mean follow-up was shorter in the biologics cohort than in the control group (3.8 vs. 7.3 years, *p* < 0.001). The proportion of biologics by class was as follows: TNF-α inhibitor, 25.3%; IL-12/23 inhibitor, 26.6%; IL-23 inhibitor, 29.8%; and IL-17 inhibitor, 18.3%. HTN prevalence was higher in the biologics cohort than in the control group (13.7% vs. 12.9%, *p* = 0.041), with no significant differences in diabetes or hyperlipidemia.

### 3.2. Prevalence of Comorbid Diseases at Baseline

The prevalence of comorbid diseases associated with biologics at baseline in the biologics and control cohorts is summarized in Table 1. The biologics group showed higher prevalence of RA (aOR, 3.29; 95% CI, 2.88–3.75), IBD (aOR, 15.33; 95% CI, 11.46–20.52), TB (aOR, 1.40; 95% CI, 1.05–1.33), HBV infection (aOR, 1.41; 95% CI, 1.15–1.74), mood disorders (aOR, 1.18; 95% CI, 1.05–1.33), NMSC (aOR, 1.90; 95% CI, 1.07–3.37), and CKD (aOR, 1.75; 95% CI, 1.30–2.36) than the control group. Conversely, lower prevalence was observed for herpes zoster (aOR, 0.90; 95% CI, 0.81–1.00), lymphoma (aOR, 0.10; 95% CI, 0.01–0.72), asthma (aOR, 0.79; 95% CI, 0.71–0.88), and COPD (aOR, 0.69; 95% CI, 0.52–0.92).

### 3.3. Incidence and Risk of Comorbid Diseases Following the Index Date

The incidence of comorbidities occurring after the index date in each cohort was analyzed. The incidence rates per 100,000 person-years and aHRs are summarized in Figure 2. Four diseases reached statistical significance in the analysis: RA, mood disorder, solid tumor, and CKD. In the biologics cohort, RA (aHR, 0.37; 95% CI, 0.17–0.79), mood disorder (aHR, 0.72; 95% CI, 0.53–0.97), and solid tumor (aHR, 0.63; 95% CI, 0.47–0.84) showed a significant decrease in risk. In contrast, CKD (aHR, 1.52; 95% CI, 1.09–2.13) demonstrated a significant increase in risk in the biologics cohort. No significant differences were observed in terms of the incidence of CV disease, including MACE, or infectious diseases, between the two cohorts. Cumulative incidence plots are shown in Appendix A.

### 3.4. Subgroup Analysis

Subgroup analysis based on the class of biologics revealed distinct patterns (Figure 3). TNF-α inhibitor use was associated with increased risks of MI (aHR, 1.86; 95% CI, 1.03–3.35), TB (aHR, 2.68; 95% CI, 1.47–4.90), and CKD (aHR, 2.27; 95% CI, 1.51–3.42), alongside a reduced risk of stroke (aHR, 0.51; 95% CI, 0.27–1.00). Moreover, the IL-12/23 inhibitor cohort exhibited no significant differences compared to the conventional systemic immunosuppressant cohort. Notably, IL-23 inhibitors reduced risks of herpes zoster (aHR, 0.47; 95% CI, 0.26–0.84) and solid tumor (aHR, 0.31; 95% CI, 0.12–0.83). Finally, IL-17 inhibitors increased risks of atherosclerosis (aHR, 3.70; 95% CI, 1.14–11.97) and COPD (aHR, 2.96; 95% CI, 1.08–8.14) but reduced those of herpes zoster (aHR, 0.49; 95% CI, 0.25–0.99) and mood disorder risks (aHR, 0.23; 95% CI, 0.06–0.94).

Additionally, a subgroup analysis was conducted based on whether the CV risk was greater than or equal to 2 or less than 2 with respect to TNF-α and IL blockers (Figure 4). In CV risk subgroups, TNF-α inhibitors were associated with a reduced risk of stroke (aHR, 0.36; 95% CI, 0.14–0.98) in the high-risk group, accompanied by an increased risk of HCV infection (aHR, 7.51; 95% CI, 1.50–37.56) and CKD (aHR, 2.08; 95% CI, 1.21–3.58). Moreover, in the low-risk group, there was a significant increase in the risks of TB (aHR, 3.75; 95% CI, 1.46–9.65) and herpes zoster (aHR, 1.52; 95% CI, 1.10–2.11). Conversely, IL inhibitors did not lead to significant differences in the risk of CV disease in both groups. However, in the high-risk group, the risks of RA (aHR, 0.14; 95% CI, 0.02–0.99) and herpes zoster (aHR, 0.53; 95% CI, 0.35–0.82) were significantly decreased, while in the low-risk group, a significant reduction was observed in the risk of solid tumors (aHR, 0.29; 95% CI, 0.12–0.70).

We additionally conducted a head-to-head comparison of comorbid disease risks among IL inhibitors (Appendix A). Stroke risk was significantly lower with IL-17 inhibitors than with IL-23 inhibitors (aHR 0.23; 95% CI, 0.06–0.90) but higher with IL-23 inhibitors than with IL-12/23 inhibitors (aHR 3.71; 95% CI, 1.05–13.11).

## 4. Discussion

This retrospective cohort study identified the prevalence and risk of comorbid diseases in patients with PsO or PsA treated with biologics, compared to those of patients treated with conventional systemic immunosuppressants. In the subgroup analyses stratified according to the class of biologics and CV risk factors, several distinct risks that were not observed in the main analysis were identified.

At baseline, certain comorbidities—including RA, IBD, TB, HBV infection, mood disorders, NMSC, and CKD—were more prevalent in the biologics cohort than in the control group. Conversely, immunobullous disease, herpes zoster, lymphoma, asthma, and COPD were less prevalent. These discrepancies likely reflect underlying disease severity, as biologics are typically prescribed only after failure of at least two prior systemic treatments (CsA, MTX, acitretin, or phototherapy) in patients with severe PsO, as stipulated by Korean reimbursement policy. In contrast, the control group likely includes patients with moderate-to-severe PsO who never escalated to biologics. Given that severe PsO is associated with a higher risk of systemic comorbidities [22,27], baseline differences in prevalence likely mirror differences in disease burden rather than treatment effect alone. In addition, the increased baseline prevalence of TB, HBV infection, and CKD in the biologics group may be explained by routine pre-biologic evaluations, which include blood tests, renal panels (e.g., creatinine and eGFR), and mandatory screening for latent infections. These assessments increase the likelihood of detecting asymptomatic comorbidities. Furthermore, patients with contraindications to conventional agents—such as renal or hepatic impairment—may be preferentially selected to undergo treatment with biologics under Korean reimbursement criteria, further increasing the proportion of individuals with these conditions in the biologics group.

Despite these baseline imbalances, the risk of several comorbidities—particularly RA, mood disorders, and solid tumors—was significantly reduced during follow-up in the biologics cohort. These findings suggest that biologics may offer long-term benefits in mitigating systemic inflammation, as well as psychological stress, and its downstream comorbidities. Considering that TNF-α, IL-17, and IL-23 are also involved in the pathogenesis of RA and IBD, this result seems reasonable [28,29]. The observed reduction in mood disorders aligns with prior data linking PsO to psychological distress and suggests a potential improvement in mental health outcomes with biologics [14,30,31]. Similarly, the reduced incidence of solid tumors, primarily observed in the IL-23 inhibitor subgroup, may suggest a potential association. However, this finding should be interpreted with caution, given the relatively short follow-up duration for IL-23 inhibitors, which were only recently approved for use in Korea. Thus, the observed pattern could reflect the limited time at risk rather than any true protective effect. While previous studies have proposed antitumor activity associated with IL-12 [32,33], further research with longer follow-up and latency-sensitive designs is needed to clarify the clinical significance of this observation. On the other hand, CKD was the only condition for which both baseline prevalence and incident risk were higher in the biologics cohort. This may reflect the severity of PsO in the included patients and cumulative nephrotoxicity from prior long-term CsA use before initiating biologics [2,34], rather than a direct adverse effect of biologics themselves.

This study primarily aimed to examine the impact of biologics on CV diseases, including MACE, among comorbid conditions. Although the main analysis did not reveal significant differences in major CV outcomes, subgroup analyses provided nuanced insights. Our findings align with those of recent large-scale cohort studies supporting the CV safety of biologics in psoriatic disease. For example, a nationwide Korean study by Song et al. reported significantly lower risks of both new-onset and recurrent MACEs in biologics users, even among patients with a history of CV events [35]. Similarly, in a prospective UK registry-based cohort, Rungapiromnan et al. observed no increased CV risk across different biologics and MTX [36]. In patients treated with TNF-α inhibitors, we observed an increased risk of MI but a decreased risk of stroke, particularly in high CV-risk individuals. While these opposing trends may reflect differential effects on vascular beds or inflammatory pathways, further mechanistic research is needed. Notably, TNF receptor 2 has been implicated in cardioprotection [37,38,39], yet its role in MI remains controversial. In contrast, among IL inhibitors, IL-23 inhibitors were associated with the highest stroke risk, while IL-17 inhibitors showed the lowest. Given the dual pro- and anti-atherogenic roles of IL-23 and IL-17 [40,41,42], the net clinical effects may depend on complex cytokine interactions.

The clinical implications of this study are that, considering the current trend of using IL inhibitors over TNF-α inhibitors in the biologics of PsO, IL inhibitors were found to have no significant impact on most CV diseases, including MACE. Among the classes of biologics, IL-23 inhibitors may be considered as a priority for patients with severe PsO accompanied by solid tumors such as lung cancer. IL-17 inhibitors, however, could be considered for patients who need to be cautious about stroke risk, but caution may be warranted when prescribing IL-17 inhibitors to patients with preexisting pulmonary risk factors such as heavy smoking. In such cases, baseline pulmonary evaluation and lifestyle counseling (e.g., smoking cessation and weight management) may help mitigate potential risks.

This study has several limitations. First, its retrospective and observational design conducted in a single ethnic population limits generalizability and causal inference. Second, although we included only patients with a principal diagnosis of PsO or PsA (L40.x), we could not fully account for the possibility that some patients in the biologics cohort may have initiated treatment for other immune-mediated conditions such as RA or IBD, potentially influencing comorbidity profiles. Third, patients with PsA were more prevalent in the biologics group than in the control group, where treatment was limited to MTX and CsA; this may reflect underlying differences in disease spectrum between groups. Fourth, the observation period spans from 2002 to 2021, during which the use of biologics increased markedly in Korea after 2017. Consequently, patients in the biologics cohort were more likely to have been treated in recent years and thus have shorter follow-up durations than those in the conventional therapy group. This temporal imbalance introduces potential calendar-time bias and may have limited the ability of our analysis to capture long-latency outcomes such as malignancies. Although adjustments were made for baseline characteristics, residual confounding by treatment sequencing and observation time cannot be excluded. Therefore, future studies with period-matched designs and longer follow-up are warranted. Fifth, smoking status was unavailable for the full cohort and thus could not be adjusted for in the COPD subgroup analysis. Given the well-established link between smoking and COPD, this unmeasured confounder may have biased the observed association with IL-17 inhibitors. Sixth, systemic retinoids, small molecules, and targeted synthetic DMARDs were not included in the control group, which may limit representativeness. Lastly, prior treatment history (e.g., drug switching or cumulative dose) was not available, and disease severity at baseline could not be fully balanced between cohorts. Despite these limitations, this study remains notable for its large sample size and real-world relevance, identifying differential comorbidity risks across biologics classes and CV risk strata in a national cohort.

## 5. Conclusions

In conclusion, this nationwide cohort study in South Korea indicates that the use of biologics in individuals with PsO or PsA is relatively safe for CV disease and reduces the risk of RA, mood disorders, and solid tumors. These findings underscore the importance of individualized treatment decisions based on comorbidity profiles, biologics class, and patient risk factors.

## Figures and Tables

**Figure 1 biomedicines-13-02219-f001:**
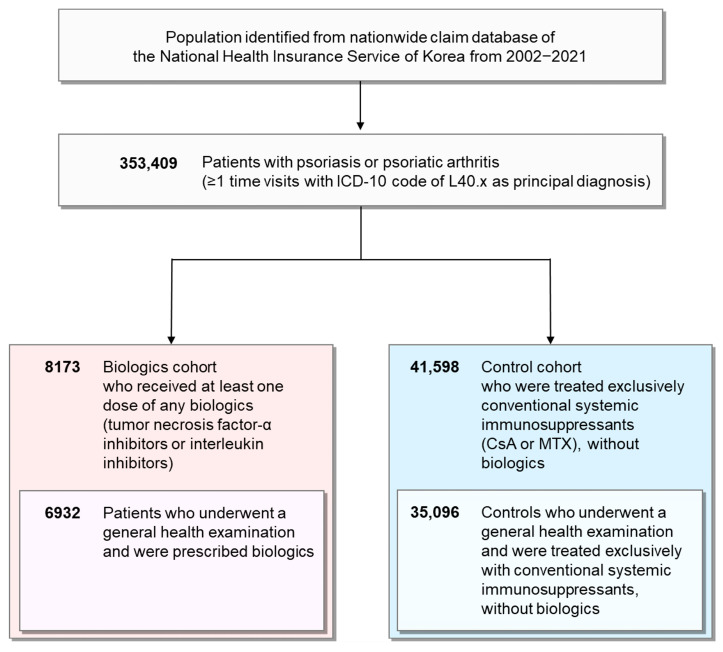
Flowchart of study population selection. We identified 353,409 individuals with psoriasis or psoriatic arthritis from the National Health Insurance Database of Korea. A biologics cohort of 8173 patients who were prescribed any biologics at least once and a control cohort of 41,598 patients who were treated exclusively with conventional systemic immunosuppressants without biologics were selected. Among them, 6932 patients in the biologics cohort and 35,096 patients in the control cohort underwent a general health examination.

**Figure 2 biomedicines-13-02219-f002:**
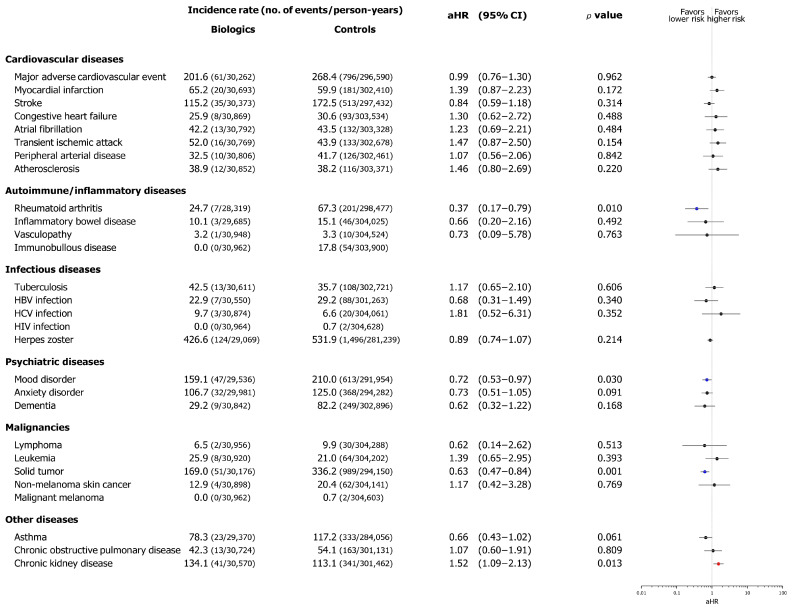
Comorbid disease risk associated with biologics in the biologics cohort compared with that in the control cohort. The forest plot depicts adjusted hazard ratios (aHRs) and 95% confidence intervals (CIs) of comorbid diseases in the biologics and control cohorts. Statistical estimates were adjusted for age, sex, insurance type, income level, and location. Abbreviations: aHRs, adjusted hazard ratios; CI, confidence interval.

**Figure 3 biomedicines-13-02219-f003:**
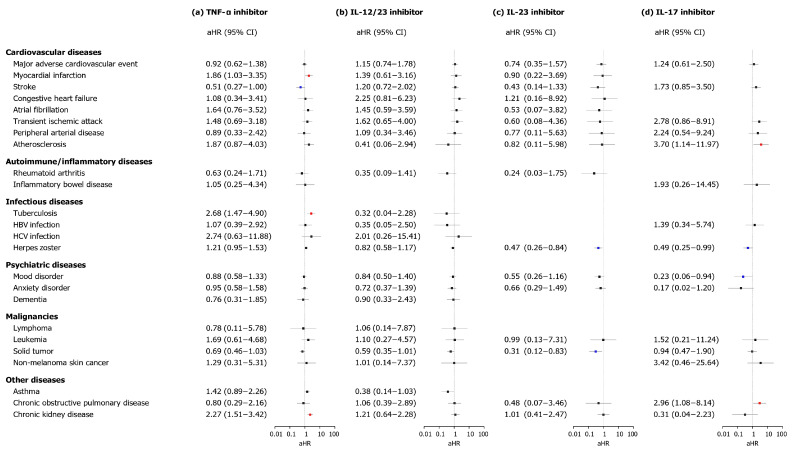
Stratified analysis of comorbid disease risks in biologics cohorts by biologic class. The forest plot depicts adjusted hazard ratios (aHRs) and 95% confidence intervals (CIs) of comorbid diseases in the biologics and control cohorts. Statistical estimates were adjusted for age, sex, insurance type, income level, and location. The risks of comorbid disease outcomes were stratified by class of biologics [(**a**) TNF-α inhibitor, (**b**) IL-12/23 inhibitor, (**c**) IL-23 inhibitor, and (**d**) IL-17 inhibitor]. Abbreviations: aHRs, adjusted hazard ratios; CI, confidence interval; IL, interleukin; TNF, tumor necrosis factor.

**Figure 4 biomedicines-13-02219-f004:**
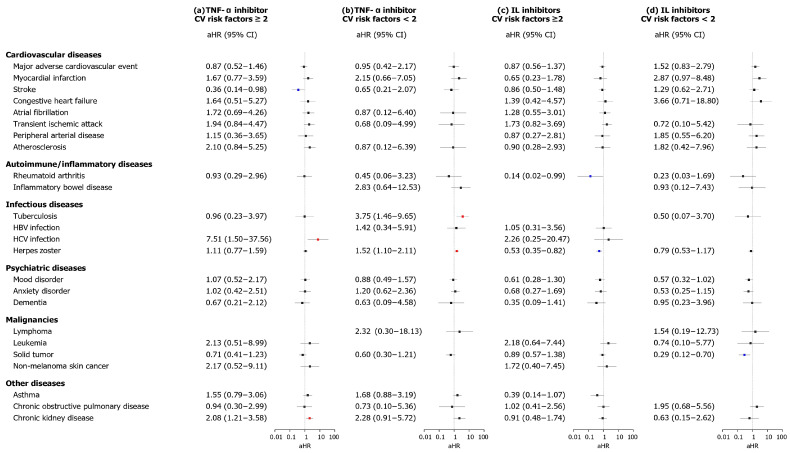
Subgroup analyses according to class of biologics and cardiovascular risk. Subgroup analyses were conducted to investigate potential differences in terms of class of biologics and CV risk: (**a**) TNF-α inhibitor cohort of the high CV risk group, (**b**) TNF-α inhibitor cohort of the low CV risk group, (**c**) IL inhibitor cohort of the high CV risk group, and (**d**) IL inhibitor cohort of the low CV risk group. The forest plot shows the aHRs and 95% CIs of comorbid disease outcomes in the biologics and control cohorts. Statistical estimates were adjusted for age, sex, insurance type, income level, and location. Abbreviations: aHRs, adjusted hazard ratios; CI, confidence interval; CV, cardiovascular; IL, interleukin; TNF, tumor necrosis factor.

**Table 1 biomedicines-13-02219-t001:** Prevalence of comorbid diseases at baseline in the biologics and control cohorts.

	Prevalent Diseases, No. (%)					
Comorbid Disease	Biologics (*n* = 8173)		Controls (*n* = 41,598)		aORs (95% CI)	
**Cardiovascular diseases**						
MACE	147	(1.80%)	738	(1.77%)	1.01	(0.85–1.21)
Myocardial infarction	39	(0.48%)	204	(0.49%)	0.97	(0.69–1.37)
Stroke	140	(1.71%)	738	(1.77%)	0.96	(0.80–1.16)
Congestive heart failure	23	(0.28%)	114	(0.27%)	1.03	(0.66–1.61)
Atrial fibrillation	33	(0.40%)	117	(0.28%)	1.44	(0.98–2.12)
Transient ischemic attack	38	(0.46%)	207	(0.50%)	0.93	(0.66–1.32)
Peripheral arterial disease	38	(0.46%)	219	(0.53%)	0.88	(0.62–1.25)
Atherosclerosis	25	(0.31%)	104	(0.25%)	1.22	(0.79–1.90)
**Autoimmune/inflammatory diseases**						
Rheumatoid arthritis	372	(4.55%)	595	(1.43%)	**3.29**	(2.88–3.75)
Inflammatory bowel disease	180	(2.20%)	61	(0.15%)	**15.33**	(11.46–20.52)
Vasculopathy	3	(0.04%)	15	(0.04%)	1.02	(0.29–3.52)
Immunobullous disease	1	(0.01%)	53	(0.13%)	**0.10**	(0.01–0.69)
**Infectious diseases**						
Tuberculosis	58	(0.71%)	211	(0.51%)	**1.40**	(1.05–1.33)
HBV infection	116	(1.42%)	420	(1.01%)	**1.41**	(1.15–1.74)
HCV infection	19	(0.23%)	77	(0.19%)	1.26	(0.76–2.08)
HIV infection	0	(0.00%)	2	(0.00%)	NA	
Herpes zoster	431	(5.27%)	2430	(5.84%)	**0.90**	(0.81–1.00)
**Psychiatric diseases**						
Mood disorder	338	(4.14%)	1468	(3.53%)	**1.18**	(1.05–1.33)
Anxiety disorder	227	(2.78%)	1215	(2.92%)	0.95	(0.82–1.10)
Dementia	25	(0.31%)	143	(0.34%)	0.89	(0.58–1.36)
**Malignancies**						
Lymphoma	1	(0.01%)	51	(0.12%)	**0.10**	(0.01–0.72)
Leukemia	7	(0.09%)	34	(0.08%)	1.05	(0.46–2.36)
Solid tumor	192	(2.35%)	988	(2.38%)	0.99	(0.85–1.16)
Non-melanoma skin cancer	16	(0.20%)	43	(0.10%)	**1.90**	(1.07–3.37)
Malignant melanoma	1	(0.01%)	2	(0.00%)	2.55	(0.23–28.07)
**Other diseases**						
Asthma	416	(5.09%)	2641	(6.35%)	**0.79**	(0.71–0.88)
COPD	56	(0.69%)	410	(0.99%)	**0.69**	(0.52–0.92)
Chronic kidney disease	58	(0.71%)	169	(0.41%)	**1.75**	(1.30–2.36)

Abbreviations: aORs, adjusted odds ratios; CI, confidence interval; COPD, chronic obstructive pulmonary disease; HBV, hepatitis B virus; HCV, hepatitis C virus; HIV, human immunodeficiency virus; MACE, major adverse cardiovascular event; NA, not calculable; No., number. Statistically significant (*p*-value < 0.05) values are highlighted in bold.

## Data Availability

The data that support the findings of this study are available from the National Health Insurance Service of Korea, but restrictions apply to the availability of the data, which were used under license for this study and are therefore not publicly available. Data can be provided by the authors with permission of the National Health Insurance Service of Korea, as determined by their review process.

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
