# Peer review of "Impact of Biologics on Comorbidities in Patients with Psoriasis or Psoriatic Arthritis"

_biomedicines, 2025, doi:10.3390/biomedicines13092219_

Round 1

Reviewer 1 Report

Comments and Suggestions for Authors

In the present research article "Impact of biologics on comorbidities in patients with psoriasis or psoriatic arthritis", Lee et al have attempted to investigate the use of biologics such as inhibitors of  TNF-α, IL-12/23, IL-23, or IL- 21 17 in patients with psoriasis or psoriatic arthritis. The authors have collected data related to patients treated with biologics and conventional immunosuppressants and have investigated the effect of biologics on the comorbidities and found that the biologics are safer to use in cardiovascular comorbidities and reduced the risk of rheumatoid arthritis, mood swing and solid tumors.

Although, study is interesting, the article requires improvement in introduction section.

The authors need to add more information in Introduction section. Introduction is section very short and there is not enough information regarding psoriasis or psoriatic arthritis and its occurrence rate in population. Further, authors should also highlight briefly about the comorbiditeis associated with psoriasis or psoriatic arthritis.

Author Response

Reviewer #1:

In the present research article "Impact of biologics on comorbidities in patients with psoriasis or psoriatic arthritis", Lee et al have attempted to investigate the use of biologics such as inhibitors of TNF-α, IL-12/23, IL-23, or IL- 21 17 in patients with psoriasis or psoriatic arthritis. The authors have collected data related to patients treated with biologics and conventional immunosuppressants and have investigated the effect of biologics on the comorbidities and found that the biologics are safer to use in cardiovascular comorbidities and reduced the risk of rheumatoid arthritis, mood swing and solid tumors.

Although, study is interesting, the article requires improvement in introduction section.

The authors need to add more information in Introduction section. Introduction is section very short and there is not enough information regarding psoriasis or psoriatic arthritis and its occurrence rate in population. Further, authors should also highlight briefly about the comorbiditeis associated with psoriasis or psoriatic arthritis.

Response to Comment: We would like to thank Reviewer 1 for your time and efforts in reviewing our manuscript and for providing comments, which have considerably helped us improve our manuscript. We have made revisions based on your comments and have provided our responses below. We hope that our responses and revisions appropriately address your comments.

In response to your suggestion, we have now expanded the Introduction section to provide additional context regarding the epidemiology and systemic nature of psoriasis and psoriatic arthritis, including their prevalence in South Korea and worldwide, as well as a brief overview of the associated comorbidities. The revised paragraph now reads:

(Page 1–2): Psoriasis (PsO) and psoriatic arthritis (PsA) are chronic, immune-mediated, multisystem diseases characterized by erythematous and scaly skin lesions and joint inflammation, respectively [1-3]. PsO affects approximately 2–3% of the global population, while PsA develops in up to 30% of individuals with PsO [2,4,5]. In South Korea, the prevalence of PsO is estimated at approximately 0.5%, and recent trends indicate a steady increase in its incidence [6]. The systemic inflammatory nature of these diseases often predisposes patients to a wide range of comorbidities [3,7,8]. Multiple studies have reported increased risks of cardiovascular (CV) disease, metabolic syndrome, chronic kidney disease (CKD), inflammatory bowel disease (IBD), rheumatoid arthritis (RA), depression, and some malignancies among individuals with PsO and PsA [9-12]. These comorbidities significantly affect overall morbidity and quality of life [13].

Reviewer 2 Report

Comments and Suggestions for Authors

The authors use a nationwide Korean NHIS claims cohort (2002–2021) to compare incident comorbidities in psoriasis/psoriatic arthritis patients treated with biologics vs. conventional systemic agents (cyclosporine or methotrexate). The authors address a clinically relevant question with a powerful dataset and provide useful descriptive patterns by biologic class. However, the current analysis is vulnerable to confounding by indication/severity and calendar-time bias, as the authors have addressed in the limitations. While the study provides potentially important insights, several revisions are necessary to address the following concerns.

1. At baseline, the biologics cohort had higher prevalence of several coded comorbidities (RA, IBD, TB, HBV infection, mood disorders, NMSC, and CKD) than the control cohort. This pattern likely reflects (a) ascertainment from the routine pre-biologic work-up (CBC, CMP including creatinine/eGFR, and infectious disease screening for TB/HBV/HCV/HIV), which preferentially detects latent infections in candidates for biologics, and (b) channeling by reimbursement policy in Korea, whereby patients may start biologics after failing prior systemic agents or when conventional agents are contraindicated (e.g., CKD or liver disease), effectively enriching the biologics cohort for these conditions. Please consider adding baseline assessment procedures done in clinics before the start of biologics and reimbursement criteria, and discuss how these factors inflate baseline prevalence differences (especially TB, HBV, and CKD).

2. Exposure is defined as ≥1 biologic prescription with time-zero at first biologic, whereas controls are indexed at first cyclosporine/methotrexate prescription. Because biologics are initiated later in the disease course (often after prior systemic therapy failure) and later in calendar time, this design induces confounding by severity/line of therapy and calendar-time bias. Recognizing that a wholesale redesign may not be feasible, I suggest pragmatic mitigations: (a) restrict the index period to 2010–2021 to align with the established biologic era in Korea (e.g., the first TNF inhibitor, etanercept, was approved around 2006, followed by infliximab in 2009. Ustekinumab was approved around 2011, and secukinumab around 2015), thereby reducing pre-biologic era heterogeneity, (b) alternatively (or additionally), report calendar-period-stratified analyses (e.g., 2002–2009, 2010–2014, 2015–2021) and show whether hazard ratios are stable across periods.

3. The authors report malignancy risk using incidence rates (IRs) per person-years, which appropriately accounts for differences in follow-up duration across cohorts, and conclude that biologics may reduce the risk of solid tumors. However, I would caution that this approach does not fully address the issue of latency for malignancies. Since many cancers have long induction and latency periods, studies with relatively short follow-up windows may underestimate the true risk, even when expressed per person-years. I recommend that the authors discuss this limitation explicitly in the manuscript and, if possible, provide additional analyses such as stratification by follow-up duration or use of lag periods to better capture delayed risk. 

Likewise, the finding that IL-23 inhibitors were associated with a reduced risk of solid tumors should be interpreted with caution. As IL-23 inhibitors were approved much later than other biologics (guselkumab 2018; risankizumab 2019 compared to ustekinumab 2011; secukinumab 2015; ixekizumab 2017 in Korea), the duration of follow-up for this cohort is substantially shorter. Given the long latency of solid tumors, the lower incidence observed may reflect insufficient time at risk rather than a true reduction in cancer risk. I recommend that the authors discuss this limitation explicitly and, if possible, provide sensitivity analyses stratified by follow-up duration.

4. The authors state that caution may be warranted when prescribing IL-17 inhibitors to patients with preexisting pulmonary risk factors, such as heavy smoking, as IL-17 inhibitors increase the risk of COPD. However, for outcomes such as COPD, baseline smoking status represents a key confounder. If smoking prevalence differed between the biologics and control cohorts and/or across biologic agents, this could have substantially influenced the observed COPD incidence rates. Please clarify whether smoking status was available and adjusted for in the analysis, or otherwise acknowledge this as a limitation.

5. I recommend that the authors include discussion of two relevant cohort studies by Song et al., (doi: 10.1016/j.jaad.2025.03.055) and Rungapiromnan et al., (doi: 10.1111/jdv.16018) as these articles provide complementary real-world and prospective evidence on the association between biologic therapies and cardiovascular outcomes in psoriasis and psoriatic arthritis, which would strengthen the context and interpretation of the current findings.

Author Response

Reviewer #2: 

The authors use a nationwide Korean NHIS claims cohort (2002–2021) to compare incident comorbidities in psoriasis/psoriatic arthritis patients treated with biologics vs. conventional systemic agents (cyclosporine or methotrexate). The authors address a clinically relevant question with a powerful dataset and provide useful descriptive patterns by biologic class. However, the current analysis is vulnerable to confounding by indication/severity and calendar-time bias, as the authors have addressed in the limitations. While the study provides potentially important insights, several revisions are necessary to address the following concerns.

  1. At baseline, the biologics cohort had higher prevalence of several coded comorbidities (RA, IBD, TB, HBV infection, mood disorders, NMSC, and CKD) than the control cohort. This pattern likely reflects (a) ascertainment from the routine pre-biologic work-up (CBC, CMP including creatinine/eGFR, and infectious disease screening for TB/HBV/HCV/HIV), which preferentially detects latent infections in candidates for biologics, and (b) channeling by reimbursement policy in Korea, whereby patients may start biologics after failing prior systemic agents or when conventional agents are contraindicated (e.g., CKD or liver disease), effectively enriching the biologics cohort for these conditions. Please consider adding baseline assessment procedures done in clinics before the start of biologics and reimbursement criteria, and discuss how these factors inflate baseline prevalence differences (especially TB, HBV, and CKD).

Response to Comment 1: We would like to thank Reviewer 2 for your time and efforts in reviewing our manuscript and for providing comments, which have considerably helped us improve our manuscript. We have made revisions based on your comments and have provided our point-by-point responses below. We hope that our responses and revisions appropriately address your comments.

We agree that baseline differences in comorbidity prevalence could be influenced by several structural and procedural factors, such as pre-biologic screening protocols and national reimbursement criteria. In response to your comment, we have now revised the Discussion section to more clearly explain how routine clinical assessments and channeling bias could have contributed to the observed baseline imbalances, as follows:

(Pages 8–9): Given that severe PsO is associated with a higher risk of systemic comorbidities [22,27], baseline differences in prevalence likely mirror differences in disease burden rather than treatment effect alone. In addition, the increased baseline prevalence of TB, HBV infection, and CKD in the biologics group may be explained by routine pre-biologic evaluations, which include blood tests, renal panels (e.g., creatinine, eGFR), and mandatory screening for latent infections. These assessments increase the likelihood of detecting asymptomatic comorbidities. Furthermore, patients with contraindications to conventional agents—such as renal or hepatic impairment—may be preferentially selected to undergo treatment with biologics under Korean reimbursement criteria, further increasing the proportion of individuals with these conditions in the biologics group.

  1. Exposure is defined as ≥1 biologic prescription with time-zero at first biologic, whereas controls are indexed at first cyclosporine/methotrexate prescription. Because biologics are initiated later in the disease course (often after prior systemic therapy failure) and later in calendar time, this design induces confounding by severity/line of therapy and calendar-time bias. Recognizing that a wholesale redesign may not be feasible, I suggest pragmatic mitigations: (a) restrict the index period to 2010–2021 to align with the established biologic era in Korea (e.g., the first TNF inhibitor, etanercept, was approved around 2006, followed by infliximab in 2009. Ustekinumab was approved around 2011, and secukinumab around 2015), thereby reducing pre-biologic era heterogeneity, (b) alternatively (or additionally), report calendar-period-stratified analyses (e.g., 2002–2009, 2010–2014, 2015–2021) and show whether hazard ratios are stable across periods.

Response to Comment 2: We appreciate the reviewer’s insightful comment regarding the potential bias introduced by differences in treatment line and calendar time. We fully acknowledge that biologics were generally introduced later, both in the disease course and in calendar time, which may have contributed to confounding by severity and temporal bias.

Unfortunately, we are currently unable to perform additional stratified analyses or redesign the index period due to the expiration of our access to the National Health Insurance Service (NHIS) database. Nevertheless, to address the reviewer’s concern, we have clarified these limitations in the revised limitations section and have explicitly acknowledged the possibility of calendar-time bias due to the longer observation period in the conventional therapy group due to the relatively more recent uptake of biologics in Korea.

We hope that these clarifications, along with the large sample size and multivariable adjustments of our analysis, will help readers interpret our findings within the appropriate methodological context. The limitations have been described below:

(Page 10): Fourth, the observation period spans from 2002 to 2021, during which the use of biologics increased markedly in Korea after 2017. Consequently, patients in the biologics cohort were more likely to have been treated in recent years and thus have shorter follow-up durations than those in the conventional therapy group. This temporal imbalance introduces potential calendar-time bias and may have limited the ability of our analysis to capture long-latency outcomes, such as malignancies. Although adjustments were made for baseline characteristics, residual confounding by treatment sequencing and observation time cannot be excluded. Therefore, future studies with period-matched designs and longer follow-up are warranted.

  1. The authors report malignancy risk using incidence rates (IRs) per person-years, which appropriately accounts for differences in follow-up duration across cohorts, and conclude that biologics may reduce the risk of solid tumors. However, I would caution that this approach does not fully address the issue of latency for malignancies. Since many cancers have long induction and latency periods, studies with relatively short follow-up windows may underestimate the true risk, even when expressed per person-years. I recommend that the authors discuss this limitation explicitly in the manuscript and, if possible, provide additional analyses such as stratification by follow-up duration or use of lag periods to better capture delayed risk.
    Likewise, the finding that IL-23 inhibitors were associated with a reduced risk of solid tumors should be interpreted with caution. As IL-23 inhibitors were approved much later than other biologics (guselkumab 2018; risankizumab 2019 compared to ustekinumab 2011; secukinumab 2015; ixekizumab 2017 in Korea), the duration of follow-up for this cohort is substantially shorter. Given the long latency of solid tumors, the lower incidence observed may reflect insufficient time at risk rather than a true reduction in cancer risk. I recommend that the authors discuss this limitation explicitly and, if possible, provide sensitivity analyses stratified by follow-up duration.

Response to Comment 3: We completely agree that the relatively short follow-up duration for patients treated with IL-23 inhibitors may not be sufficient to fully capture the long latency period of solid tumors; thus, the observed reduced risk should be interpreted with caution.

To address this concern, we have now revised the Discussion section to temper our interpretation of this finding. Specifically, we now acknowledge that the lower incidence of solid tumors in the IL-23 inhibitor cohort may reflect the shorter time at risk rather than any true protective effect. We have also mentioned that IL-23 inhibitors were only recently approved in Korea, thus limiting the follow-up period available for analysis.

As our NHIS database access period has expired, we were unable to conduct additional stratified analyses by follow-up duration or implement any lag-time sensitivity analyses. However, we have clearly acknowledged this limitation in the revised manuscript and will highlight the need for further studies with extended follow-up to more accurately assess the risk of malignancy associated with different classes of biologics.

We sincerely appreciate the reviewer’s thoughtful recommendations, which have helped us improve the clarity and scientific rigor of our interpretation. The revised paragraph reads as follows:

(Page 9): Similarly, the reduced incidence of solid tumors, primarily observed in the IL-23 inhibitor subgroup, may suggest a potential association. However, this finding should be interpreted with caution, given the relatively short follow-up duration for IL-23 inhibitors, which were only recently approved for use in Korea. Thus, the observed pattern could reflect the limited time at risk rather than any true protective effect. While previous studies have proposed anti-tumor activity associated with IL-12 [32,33], further research with longer follow-up and latency-sensitive designs is needed to clarify the clinical significance of this observation.

  1. The authors state that caution may be warranted when prescribing IL-17 inhibitors to patients with preexisting pulmonary risk factors, such as heavy smoking, as IL-17 inhibitors increase the risk of COPD. However, for outcomes such as COPD, baseline smoking status represents a key confounder. If smoking prevalence differed between the biologics and control cohorts and/or across biologic agents, this could have substantially influenced the observed COPD incidence rates. Please clarify whether smoking status was available and adjusted for in the analysis, or otherwise acknowledge this as a limitation.

Response to Comment 4: We greatly appreciate the reviewer’s insightful comment regarding smoking status as a critical confounder of COPD outcomes. As noted, the subgroup analysis assessing COPD risk by biologic class was conducted using the entire cohort, many of whom lacked smoking data. Smoking information was only available for a subset of patients who underwent general health examination. Consequently, we could not adjust for smoking status in the analysis of COPD risk.

We fully agree that differences in baseline smoking prevalence across different treatment groups may have influenced the observed association between IL-17 inhibitors and COPD, which represents an important limitation of our analysis.

To address this, we have now revised the limitations section as follows:

(Pages 10): Fifth, smoking status was unavailable for the full cohort and thus could not be adjusted for in the COPD subgroup analysis. Given the well-established link between smoking and COPD, this unmeasured confounder may have biased the observed association with IL-17 inhibitors.

  1. I recommend that the authors include discussion of two relevant cohort studies by Song et al., (doi: 10.1016/j.jaad.2025.03.055) and Rungapiromnan et al., (doi: 10.1111/jdv.16018) as these articles provide complementary real-world and prospective evidence on the association between biologic therapies and cardiovascular outcomes in psoriasis and psoriatic arthritis, which would strengthen the context and interpretation of the current findings.

Response to Comment 5: As recommended, we have now added a summary of the findings from the studies by Song et al. and Rungapiromnan et al. to strengthen the context of our cardiovascular risk findings in psoriatic disease. The relevant text has been incorporated into the Discussion section as follows:

(Page 9): This study primarily aimed to examine the impact of biologics on CV diseases, including MACE, among comorbid conditions. Although the main analysis did not reveal significant differences in major CV outcomes, subgroup analyses provided nuanced insights. Our findings align with those of recent large-scale cohort studies supporting the CV safety of biologics in psoriatic disease. For example, a nationwide Korean study by Song et al. reported significantly lower risks of both new-onset and recurrent MACEs in biologics users, even among patients with a history of CV events [35]. Similarly, in a prospective UK registry-based cohort, Rungapiromnan et al. observed no increased CV risk across different biologics and MTX [36].

Round 2

Reviewer 2 Report

Comments and Suggestions for Authors

The authors have acknowledged the methodological limitations, including confounding and calendar-time bias, and revised the discussion accordingly. While it is unfortunate that additional analyses could not be performed, these limitations are now clearly stated, and the overall balance of the Discussion is improved.